# Menstrual Cycle Phases Influence on Cardiorespiratory Response to Exercise in Endurance-Trained Females

**DOI:** 10.3390/ijerph18030860

**Published:** 2021-01-20

**Authors:** Beatriz Rael, Víctor M. Alfaro-Magallanes, Nuria Romero-Parra, Eliane A. Castro, Rocío Cupeiro, Xanne A. K. Janse de Jonge, Erica A. Wehrwein, Ana B. Peinado

**Affiliations:** 1LFE Research Group, Department of Health and Human Performance, Faculty of Physical Activity and Sport Science (INEF), Universidad Politécnica de Madrid, 28040 Madrid, Spain; beanad16@gmail.com (B.R.); n.romero@upm.es (N.R.-P.); elianeaparecidacastro@gmail.com (E.A.C.); rocio.cupeiro@upm.es (R.C.); anabelen.peinado@upm.es (A.B.P.); 2Faculty of Education, Universidad Católica de la Santísima Concepción, 4090541 Concepción, Chile; 3School of Environmental and Life Sciences, University of Newcastle, Newcastle 2308, Australia; x.jansedejonge@newcastle.edu.au; 4Department of Physiology, Michigan State University, East Lansing, MI 48824, USA; wehrwei7@msu.edu

**Keywords:** sex hormones, estradiol, progesterone, eumenorrheic, high intensity interval exercise, athletes

## Abstract

The aim of this study was to analyse the impact of sex hormone fluctuations throughout the menstrual cycle on cardiorespiratory response to high-intensity interval exercise in athletes. Twenty-one eumenorrheic endurance-trained females performed an interval running protocol in three menstrual cycle phases: early-follicular phase (EFP), late-follicular phase (LFP) and mid-luteal phase (MLP). It consisted of 8 × 3-min bouts at 85% of their maximal aerobic speed with 90-s recovery at 30% of their maximal aerobic speed. To verify menstrual cycle phase, we applied a three-step method: calendar-based counting, urinary luteinizing hormone measurement and serum hormone analysis. Mixed-linear model for repeated measures showed menstrual cycle impact on ventilatory (EFP: 78.61 ± 11.09; LFP: 76.45 ± 11.37; MLP: 78.59 ± 13.43) and heart rate (EFP: 167.29 ± 11.44; LFP: 169.89 ± 10.62; MLP: 169.89 ± 11.35) response to high-intensity interval exercise (F_2.59_ = 4.300; *p* = 0.018 and F_2.61_ = 4.648; *p* = 0.013, respectively). Oxygen consumption, carbon dioxide production, respiratory exchange ratio, breathing frequency, energy expenditure, relative perceived exertion and perceived readiness were unaltered by menstrual cycle phase. Most of the cardiorespiratory variables measured appear to be impassive by menstrual cycle phases throughout a high-intensity interval exercise in endurance-trained athletes. It seems that sex hormone fluctuations throughout the menstrual cycle are not high enough to disrupt tissues’ adjustments caused by the high-intensity exercise. Nevertheless, HR based training programs should consider menstrual cycle phase.

## 1. Introduction

The natural menstrual cycle is perhaps the second most important biological rhythm, next to the circadian one [1], and it is regulated by the hypothalamic-pituitary-ovarian axis and all hormones involved in it (predominantly, follicle-stimulating hormone [FSH], luteinizing hormone [LH], 17β-estradiol [E2] and progesterone). Despite individual variations, female sex hormones fluctuate fairly predictably over 23–38 days [2], giving rise to the different phases of the menstrual cycle. The first one is the early-follicular phase (EFP), characterised by low concentrations of sex hormones, which starts at the onset of menstruation. Then, E2 starts to rise throughout the mid-follicular phase, reaching its peak in the late-follicular phase (LFP), followed by the peak in LH and FSH, just prior to ovulation. These hormones drastically decrease after ovulation whereas progesterone starts to increase, achieving its peak in the mid-luteal phase (MLP), coinciding with high levels of E2 as well. Finally, during the late luteal phase all sex hormones drop, starting the cycle again [2,3].

Female sex hormones, specially E2 and progesterone, have receptors in several tissues of the body. Thereby, other than reproductive functions, these hormones may influence many other physiological systems such as hypothalamus, cardiovascular system, kidney tubules, liver, skeletal muscle and adipose tissue [1,3,4,5], which may have an impact on females’ exercise performance. In this sense, an increase in ventilation (Ve) has been reported in sedentary [6] and active females [7,8] as well as an increase in heart rate (HR) in both, sedentary [9] and trained females [10], during the luteal phase. In addition, higher fat utilisation in the luteal phase has been observed in active females, resulting in a lower respiratory exchange ratio (RER) during this phase [4]. However, other studies concluded no impact of menstrual cycle on maximal oxygen consumption (VO_2max_), Ve, RER, lactate and HR in physically active females [11,12,13].

These conflicting findings may be explained by methodological shortcomings, mainly the measurements trials carried out in different moments of the menstrual cycle since it has been divided into two [12,13], three [8,10] or four [6,11] phases. An additional limitation is the menstrual cycle verification, as studies often rely on calendar counting [4,11,12] or measuring body basal temperature [4], and it is well known than these methods are not accurate enough and should be accompanied by urinary LH tests and serum sex hormone verification, as a recent review concluded [3]. Therefore, the aim of this investigation was to assess the influence of sex hormone fluctuations throughout the menstrual cycle on cardiorespiratory response to high intensity interval exercise. Based on previous literature, we hypothesis that cardiorespiratory response to exercise is altered by sex hormones fluctuations over the menstrual cycle in endurance-trained females.

## 2. Material and Methods

### 2.1. Participants

A total of twenty-one eumenorrheic females (age: 30.5 ± 6.5 years; height: 163.1 ± 6.4 cm; body weight: 58.4 ± 8.7 kg; body fat percentage: 25.2% ± 6.7%; lean mass, considering it as body weight minus fat mass and minus bone mineral content: 70.38% ± 6.51%; peak oxygen consumption [VO_2preak_]: 48.4 ± 4.4 mL·min^−1^·kg^−1^) participated in this study. They had regular menstrual cycle, occurring from 23 to 38 days in length during the six months prior the study [2]. Concretely, volunteers´ menstrual cycle ranged from 28 ± 2 to 31 ± 2 days in length. All of them were healthy and well-trained (7.4 ± 5.3 years of endurance training experience with a training volume of 295.9 ± 183.6 min per week during the 6 months prior to recruitment), in endurance activities such as running, obstacle races, triathlon and cycling. Participants were required to meet the following criteria: (a) healthy adult females between 18 and 40 years old; (b) presenting with healthy iron parameters (serum ferritin > 20 μg/L, haemoglobin > 115 μg/L and transferrin saturation > 16%); (c) performing endurance training between 3 and 12 h per week. Exclusion criteria included: (a) irregular menstrual cycles; (b) oral contraceptive use; (c) menopause; (d) smoking; (e) metabolic or hormonal disorder; (f) medication or dietary supplements that alter vascular function (e.g., tricyclic antidepressants, α-blockers, β-blockers, etc.); (g) any surgical interventions (e.g., ovariectomy); (h) pregnancies in the year preceding; (i) any musculoskeletal injury in the last six months. At the start of the data collection, all participants conducted a questionnaire gathering information about training experience, health status, dietary supplements and menstrual cycle aspects. All participants were informed about the procedures and risks involved and informed consent was provided by each participant. The experimental protocol was approved by the Institutional Ethics Committee and is in accordance with The Code of Ethics of the World Medical Association (Declaration of Helsinki).

### 2.2. Study Design

The present work is part of the IronFEMME study, an observational cross-sectional study performed by physically active and healthy women. The project consisted on two sections carried out at the same time: iron metabolism (Study I, which exercise protocol was an interval running test) and muscle damage (Study II, which protocol was based on a resistance exercise trial). Concretely, the present work shows data from Study I.

Participants came to our laboratory on four occasions. The initial screening visit was conducted during the EFP (i.e., between 2nd and 5th day of the menstrual cycle with day 1 being onset of menstrual bleeding). Volunteers came to our laboratory between 8 and 10 a.m. in a rested and overnight fasted state. Volunteers did not perform moderate or vigorous physical activity and did not take caffeine, alcohol or any supplementation 24 h prior to the screening day. Firstly, they signed all the informed consents and participant´s weight and height were recorded. Then, baseline blood samples were collected, for a complete blood count, genetic testing, biochemistry and hormonal analyses. Subsequently, an absorptiometry by dual-energy X-ray (DXA) was done. This screening session was completed with a maximal aerobic ramp test on a computerized treadmill (H/P/COSMOS 3PW 4.0, H/P/Cosmos Sports & Medical, Nussdorf-Traunstein, Germany) to determine their VO_2_peak. Expired gases were measured breath-by-breath with the gas analyser Jaeger Oxycon Pro (Erich Jaeger, Viasys Healthcare, Friedberg, Germany) for which validity and reliability have been previously demonstrated [14,15]. Heart response was continuously monitored with a 12-lead ECG. Participants began with a warm-up of 3 min at 6 km/h. Once the warm-up finished, the speed was set at 8 km/h and then increased by 0.2 km/h every 12 s until exhaustion. A slope of 1% was set throughout the test to simulate air resistance [8]. The maximal aerobic speed was considered as the minimum speed required to elicit the VO_2_peak [16]. To verify that VO_2_peak was reached, a confirmatory test was carried out as suggested in previous studies [17,18] after a 5 min recovery of the maximal aerobic test [18]. The speed equivalent to 85% of the maximal aerobic speed was calculated to use in the interval running protocol.

After this screening day, participants attended the laboratory to perform the interval running protocol in three different menstrual cycle phases: EFP (day 3.43 ± 0.93), LFP (day 11.95 ± 2.54), and MLP (day 21.86 ± 3.05). In addition, the average day of the positive result in the LH test was 14.02 ± 2.55. In order to avoid learning effects that could influence our results, the order of these running protocols was randomized, and in no case, an order involved evaluating a volunteer in more than two cycles: EFP-LFP-MLP; LFP-MLP-EFP; MLP-EFP-LFP; LFP-EFP-MLP; EFP-MLP-LFP.

### 2.3. Interval Running Protocol

To avoid diurnal variability [3], participants came to the laboratory between 8 and 10 a.m., after abstaining from alcohol or caffeine consumption and any intense physical activity or sport practice the 24 h prior the testing day. Nutritional recommendations were provided to the participants by a nutritionist in order to standardize the diet, and volunteers followed these 24 h prior to every test. In addition, participants replicated the same breakfast in each protocol performed in the different menstrual cycle phases. Figure 1 shows the protocol of the testing procedure day. Firstly, a blood sample was collected to analyze sex hormones, followed by a standing blood pressure (BP) measurement, using the auscultatory method with a calibrated sphygmomanometer. Subsequently, participants started the interval running protocol consisting of a 5 min warm-up at 60% of their maximal aerobic speed followed by 8 bouts of 3 min at 85% of their maximal aerobic speed with 90-s recovery at 30% of their maximal aerobic speed between bouts. Finally, 5 min cool down was performed at 30% of their maximal aerobic speed. During this protocol, Ve, breathing frequency (BF), VO_2_, carbon dioxide production (VCO_2_), RER, HR and energy expenditure (EE) were continuously measured using the same apparatus as mentioned for the maximal aerobic test. Cardiorespiratory values were obtained as the mean of the 5 min warm-up, as well as the mean of the 5 min cool down. Likewise, values over the interval running protocol were elicited as the mean of the 3 min high intensity intervals and the mean of the 90-s recovery intervals.

Additionally, rate of perceived exertion (RPE) and perceived readiness (PR) were respectively measured by RPE Borg 6–20 scale [19] and PR Nurmekivi 1–5 scale [20]. Participants were asked for RPE in the last 5 s of warm-up and every running bout, and at the end of the cool down. PR scale was applied in the last 5 s of warm-up and active recovery intervals from 1 to 7, and at the end of the cool down.

### 2.4. Menstrual Cycle Monitoring and Phase Determination

Considering the first day of the cycle the onset of menstruation, the days of testing were: between the 2nd and the 5th day of the cycle for the EFP, between one and three days before the ovulation day for the LFP and between five and nine days following ovulation for the MLP. These three specific phases were selected in order to analyse different hormonal environments as literature suggests [2,21]: low E2 and progesterone levels in the EFP, low progesterone but high E2 levels in the LFP and elevated levels of both progesterone and E2 in the MLP. In order to meet this and based on the literature [2,3,21], we applied a three-step method: calendar-based counting, urinary LH measurement and serum hormone analysis.

Firstly, participants were asked to record information about the length of their last six menstrual cycles. These data were provided to a gynaecologist, who confirmed the menstrual cycles were regular and estimated the ovulation day (the middle day of the menstrual cycle ± 1) as well as the menstrual cycle phases. Then, a hormone ovulation predictor kit (Ellatest, Alicante, Spain) was used to identify the surge of LH in urine. Second morning mid-stream urine sample was collected day to day from three to five days before LFP protocol until LH surge detection, which occurs 14–26 h before ovulation [2]. If LH surge was not detected or was detected more than 3 days after completion of LFP test, this test was discarded and the dates for the LFP test were recalculated to repeat it. Finally, serum sex hormones (LH, FSH, E2 and progesterone) were measured in each of the menstrual cycle phases selected for the study. Minimum progesterone was set at 16 nmol·L^−1^ in the MLP as a reliable indicator of an ovulatory non luteal phase-deficient cycle [13,22,23].

### 2.5. Blood Samples Analyses

To avoid diurnal variability [3], blood samples were taken at the same time for all volunteers, between 8–10 a.m. They were obtained with venipuncture into a vacutainer containing clot activator. Following inversion and clotting, the whole blood was centrifuged (LMC-3000 version V.5AD, Biosan, Riga, Latvia) for ten minutes at 3000 rpm. After that, serum was transferred into eppendorf tubes and stored frozen at −80 °C until further analysis. Within 1 to 15 days after testing, the serum samples were delivered to the clinical laboratory of the Spanish National Centre of Sport Medicine (Madrid, Spain) to determine sex hormones in order to verify hormonal profiles. Total E2, progesterone, FSH and LH were measured via ADVIA Centaur ^®^ solid-phase competitive chemiluminescent enzymatic immunoassay (IMMULITE 1000 system; Siemens Healthineers AG, Munich, Germany). Inter- and intra-assay coefficients of variation (CV) reported by the laboratory for each variable were, respectively: 11.9% and 8.5% at 93.3 pg/mL and 6.8% and 4.7% at 166 pg/mL for E2; 23.1% and 11.8% at 0.7 ng/mL and 5.2% and 2.5% at 9.48 ng/mL for progesterone, 5.3% and 1.8% at 1.2 mIU/mL for FSH and 5.2% and 1.8% at 0.54 mIU/mL for LH.

### 2.6. Statistical Analysis

Data are presented as mean, standard deviation of the mean (±SD) in tables and standard error of the mean (±SEM) in figures. A Shapiro-Wilk test to assess the normal distribution of the variables was conducted. A linear mixed model for repeated measures was used to analyze menstrual cycle phases (EFP, LFP and MLP), time of measurement (bouts and active recovery intervals) and time* menstrual cycle phase effects on performance variables (HR, VO_2_/kg, VCO_2_, RER, Ve, BF, EE, RPE, PR). However, the focus of the analysis is on changes over the menstrual cycle phases and the paper will not report changes over time within the protocol. Bonferroni post-hoc tests were conducted where significant differences were found in any of the analyzed factors. Additionally, a non-parametric Friedman ANOVA for repeated measures was performed to analyze differences in sex hormone concentrations, resting BP, warm-up and cool down variables among the menstrual cycle phases tested. A non-parametric Wilcoxon signed-rank test was performed to obtain post-hoc pairwise comparisons where significant differences were found. Effect sizes were calculated to assess the magnitude of effect in the changes found for non-parametric pairwise comparisons using coefficient r [24], while for Bonferroni post-hoc comparisons, Cohen’s d [25] were calculated to assess the magnitude of effect in the changes found. In order to unify the effect size under a sole coefficient, r values were converted to d values as proposed by Rosenthal [24]. Threshold values were set as small (≥0.2 and <0.5), moderate (≥0.5 and <0.8) and large (≥0.8) [25]. Confidence intervals (95% CI) were also calculated. Statistical significance was set at *p* < 0.05 and all procedures were conducted with SPSS software 21 version (IBM Corp., Armonk, NY, USA).

## 3. Results

Firstly, sex hormone concentrations throughout the menstrual cycle phases tested (Table 1) showed significant differences for LH, FSH, E2, progesterone and E2/progesterone ratio.

### 3.1. Baseline

At rest neither SBP (EFP: 106.15 ± 8.44, LFP: 109.00 ± 12.41 and MLP: 111.00 ± 8.97 mmHg) nor DBP (EFP: 65.75 ± 7.66, LFP: 67.75 ± 10.94 and MLP: 65.55 ± 7.56 mmHg) showed differences over the menstrual cycle phases (c^2^ = 5.344; *p* = 0.069 and c^2^ = 0.781; *p* = 0.677, respectively). In addition, no difference in initial RPE (c^2^ = 0.269; *p* = 0.874) was found among testing days (EFP: 6.95 ± 1.43, LFP: 6.95 ± 1.16 and MLP: 7.05 ± 1.32).

### 3.2. Warm-Up

Most of the measured variables reported to be steady over the different menstrual cycle phases over the warm-up (Table 2), although Ve, VO_2_/kg, EE, and PR showed significant difference among menstrual cycle phases. Specifically, lower values of Ve were found in the LFP compared to the EFP (*p* = 0.034, d = 0.85, CI = 0.56 to 1.14) and MLP (*p* = 0.001, d = −1.36, CI = −1.84 to −0.88). Moreover, both VO_2_/kg and EE exhibited lower values in the LFP than in the EFP (*p* = 0.013, d = 0.98, CI = 0.58 to 1.39 and *p* = 0.008, d = 1.05, CI = 0.70 to 1.40, respectively). However, post-hoc pairwise comparisons reported no significant differences among menstrual cycle phases for PR.

### 3.3. Interval Running Protocol

Most variables measured throughout the high intensity exercise, bouts reported to be unchanged when studying menstrual cycle phases and time* menstrual cycle interaction. Even though Ve reported a main effect of menstrual cycle phase, post-hoc pairwise comparisons did not show significant differences among menstrual cycle phases. Additionally, HR exhibited lower values in the EFP compared to the LFP (*p* = 0.016, d = 1.06, CI = 0.41 to 1.71). Menstrual cycle phase and time* menstrual cycle phase effects are shown in Figure 2.

According to the active recoveries throughout the interval running protocol, menstrual cycle phase and time* menstrual cycle interaction showed no effect on cardiorespiratory variables, except for Ve, which reported lower values in the LFP compared to the EFP (*p* = 0.019, d = −0.53, CI = −0.97 to 0.10) and the MLP (*p* = 0.019, d = −0.42, CI = −0.85 to 0.02). Figure 3 shows results regarding menstrual cycle phase and time* menstrual cycle phase effects.

### 3.4. Cool Down

Lastly, the response during the cool down (Table 3) were no different among menstrual cycle phases except for the following variables: Ve, VCO_2_, BF and EE. Concretely, Ve, VCO_2_ and EE showed lower values in the LFP than in the MLP (*p* = 0.008, d = −1.05, CI = −1.54 to −0.56; *p* = 0.013, d = −0.98, CI = −1.41 to −0.54; and *p* = 0.022, d = −0.91, CI = −1.30 to −0.52; respectively). In addition, lower values of BF were found during the EFP compared to the MLP (*p* = 0.033, d = −0.85, CI = −1.19 to −0.51).

## 4. Discussion

The hypothesis of the present investigation is that cardiorespiratory response to exercise is altered by sex hormones fluctuations across the menstrual cycle in endurance-trained females. Our hypothesis has been confirmed since the main finding was that menstrual cycle phase effect on Ve and HR when performing a high intensity interval running exercise.

The present study showed a menstrual cycle phase impact on Ve throughout the warm-up, the interval running protocol and the cool down, whereas post-hot comparisons were not statistically different. Outcomes from the present study are supported by previous research, which observed menstrual cycle effect on this variable. Specifically, elevated values of Ve during the MLP compared to the LFP [8] and to the EFP [26] were reported. Authors from these studies agree in the fact that increments in cardiorespiratory variables occur during the MLP due to progesterone´s peak in this phase. On the one hand, there is a strong basis in evidence that high levels of progesterone enhance the chemosensitivity of the hypothalamus chemoreceptors, lowering the threshold of the medullary respiratory centre, and this in turn increases Ve [1,3,8,26,27,28,29], which may be accompanied by a rise in VO_2_ [8,26]. In addition, due to the presence of progesterone receptors in the hypoglossal nuclei, this sex hormone relaxes bronchial smooth muscles and reduces respiratory muscles contractions [27], which may account for the increase in flow rate and Ve [29]. On the other hand, progesterone has been associated with increments in thermoregulatory setpoint, resulting in a rise in body basal temperature 0.3 to 0.5 °C [3]. In order to dissipate the heat, a redistribution of blood flow occurs, increasing the blood flow to skin whereas decreases the central one. Hence, an increase in HR [3,30] and Ve [6,22,26] may take place to maintain the cardiac output.

With regard to the cardiovascular system, even though females from the present study did not exhibit menstrual cycle phase impact neither in the warm-up nor in the cool down, HR reported a main effect of menstrual cycle phase throughout the high intensity intervals. Concretely, HR showed lower values in the EFP compared to the LFP. Likewise, a recent research conducted with endurance trained females reported higher values of HR during the MLP compared to the mid-follicular phase [10]. This study also reported increments in body basal temperature during this phase and, as aforementioned, it may be accompanied by an increase in HR [3,30] and Ve [6,22,26].. However, some other previous studies reported no effect of menstrual cycle phase on HR response to exercise. They suggested that the increase in cardiorespiratory strain due to high intensity exercise is greater than any possible increase caused by progesterone. Hence, progesterone effect on this physiological variable may be masked by the high intensity exercise [3,10,31]. Thus, discrepancies in results could be related with the intensity of the protocols. Studies reporting no menstrual cycle effect on cardiorespiratory response in trained females (Ve, VO_2_ and HR) were carried out with protocols such as 30 min constant load cycling at the MLSS [31], 40 min running at 75% of their maximal aerobic speed [10], 15 min incremental rowing ergometer test [13], 15 min incremental running protocol utilizing the Bruce Protocol [12] and incremental cycling test to exhaustion [11].

Moving on to sex hormones and females’ substrate metabolism, females from the present study exhibited different values of EE over the menstrual cycle phases during the warm-up and cool down. Retrospective studies strongly suggest that E2 promotes fat utilization and glycogen sparing. This sex hormone improves epinephrine and growth hormone levels, which has been associated with increments in hormone sensitive lipase secretion and, therefore, fat free acids release [4,30]. Moreover, E2 stimulates adenosine monophosphate kinase [1,3,4,5,7,30,31], leading to an increase in lipid oxidation. In addition, progesterone has been also associated with greater fat utilization [7,12] and glucose sparing [1,7,12,30]. However, outcomes from the interval running protocol revealed steady values of RER and EE when analysing sex hormones fluctuations throughout the menstrual cycle, as previous findings pointed out [11,13]. Several researchers concluded that substrate availability, training status and diet may have a greater effect on substrate metabolism than sex hormones [3,11,13]. In fact, there is a study in which no correlation was found between E2 and RER during submaximal runs neither in the MFP nor in the MLP [4]. In whole, complex physiological adjustments are required during high intensity exercise in order to meet physiological demands; so that, it seems that sex hormones fluctuations are not sufficient to disrupt these adjustments. Besides, endurance training cause adaptations which might outweigh any potential differences to sex hormones [4]. In this sense, the lack of menstrual cycle impact on substrate metabolism could be related with volunteers training status, since they were physically active females (>30 min per day, 3 days per week) [11], national and international cyclist athletes [13], recreationally trained cyclists [13] and endurance well-trained in the present study. Moreover, the absence of correlation between E2 and RER during submaximal runs was also observed in endurance athletes (40km of running per week or performing equivalent aerobic exercise such as cycling or swimming) [4].

The current study attempts to address a gap in the research through investigation of cardiorespiratory performance in well-trained females. The strengths of our study included its robust methodology, highlighting an accurate menstrual cycle verification, specific hormonal environments selected for the testing days and a homogeneous eumenorrheic group (well-trained and healthy females). Nonetheless, the present study has some limitations such us the uncontrolled of volunteers´ ethnic, daily habits, stress and motivation that may have altered our findings. It should be noted that different hormonal profiles such as menopause, postmenopause and oral contraceptive use might be also interesting to analyse.

## 5. Conclusions

The status of the current research suggests that sex hormone fluctuations throughout the menstrual cycle appear not to be high enough to disrupt physiological adjustments caused by high intensity interval exercise. However, Ve and HR seem to be the most altered variables across the menstrual cycle and, therefore, HR based training programs should consider menstrual cycle phase. Nonetheless, due to high variability in sex hormones concentrations between subjects and from day to day within subjects during any particular phase, individual considerations should be taken into account when training females. Besides, in order to enable a better understanding, further research regarding the effect of the menstrual cycle on cardiorespiratory response and adaptation to exercise is warranted.

## Figures and Tables

**Figure 1 ijerph-18-00860-f001:**
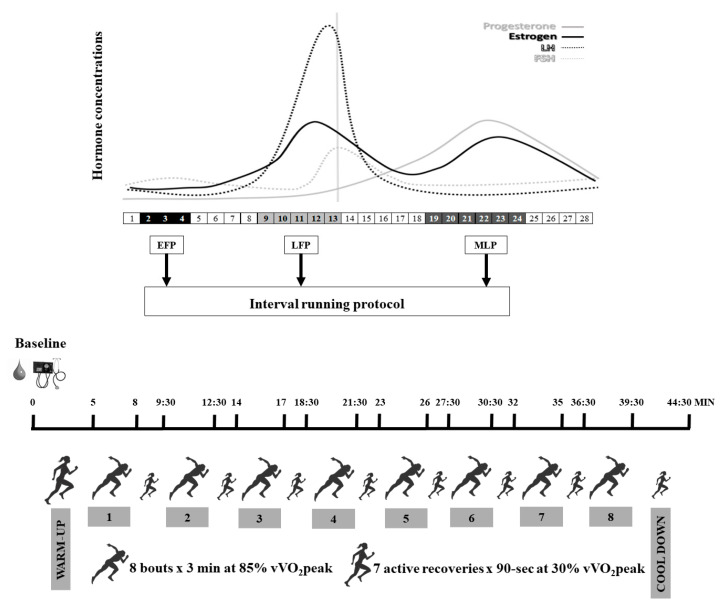
Protocol of the testing procedure day. EFP: early-follicular phase; LFP: late-follicular phase; MLP: mid-luteal phase; v VO_2_peak: maximal aerobic speed.

**Figure 2 ijerph-18-00860-f002:**
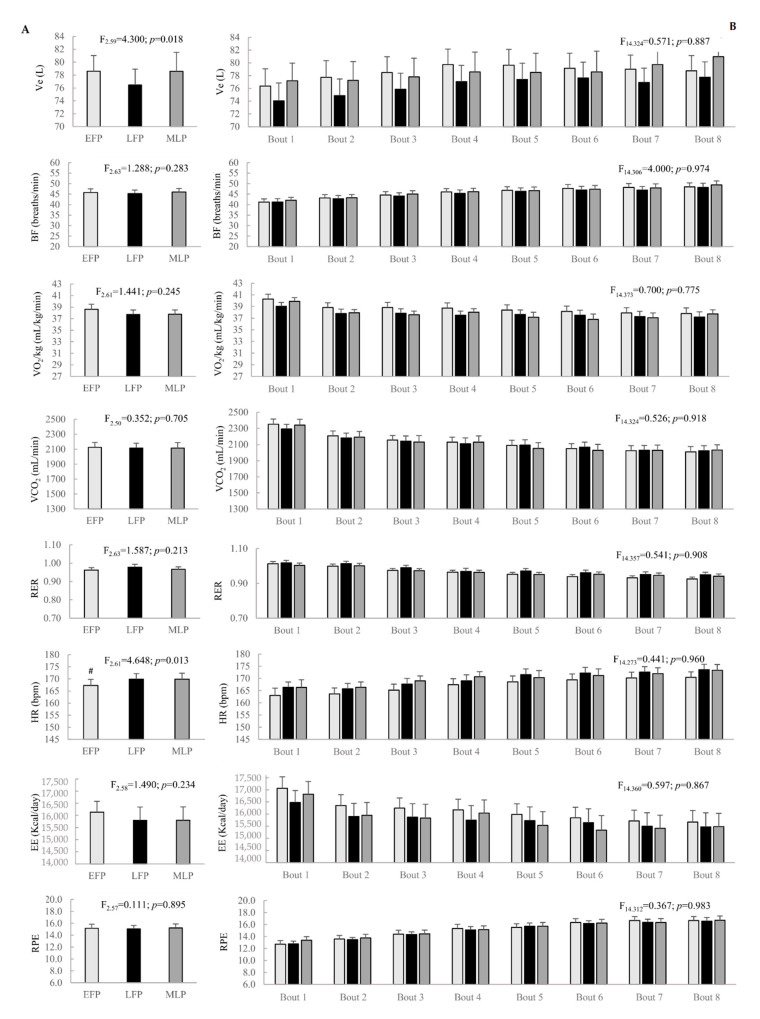
Menstrual cycle phase effect (**A**) and time* menstrual cycle interaction (**B**) on performance variables in the bouts throughout the interval running protocol. Ve: ventilation; BF: breathing frequency; VO_2_: oxygen consumption; VCO_2_: carbon dioxide production; RER: respiratory exchange ratio; HR: heart rate; EE: energy expenditure; RPE: rate of perceived exertion; EFP: early-follicular phase; LFP: late-follicular phase; MLP: mid-luteal phase. ^#^ Significant differences in LFP compared to EFP.

**Figure 3 ijerph-18-00860-f003:**
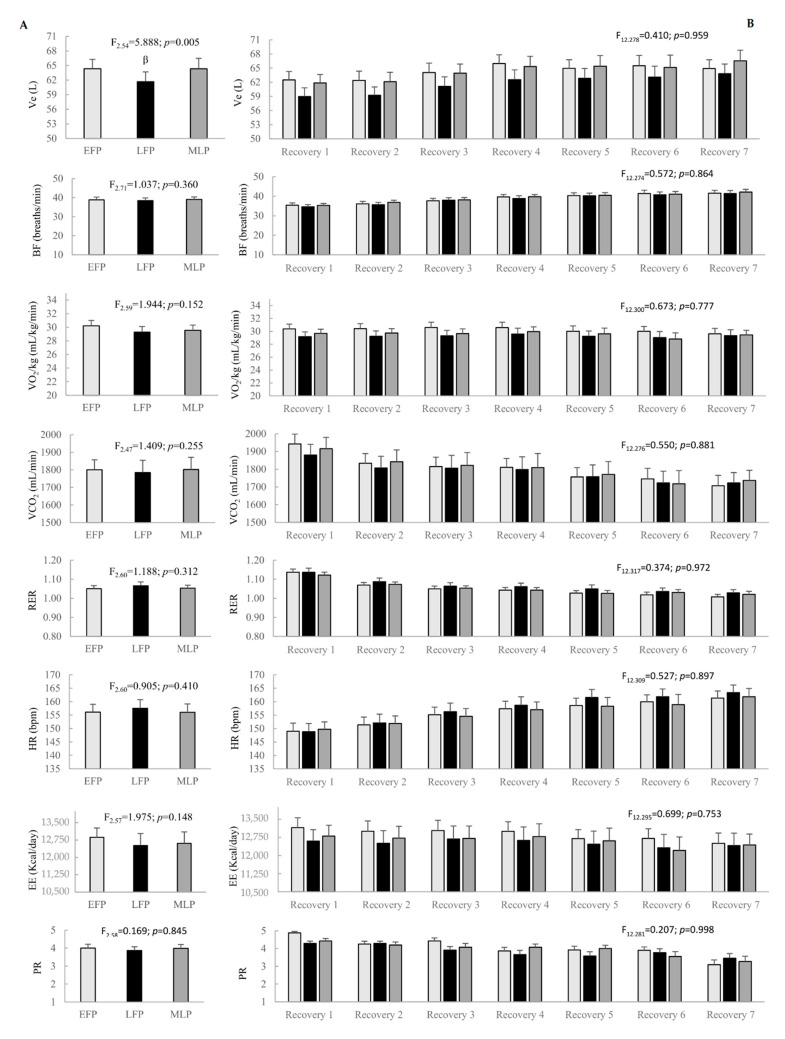
Menstrual cycle phase effect (**A**) and time* menstrual cycle phase interaction (**B**) on performance variables in the active recoveries throughout the interval running protocol. Ve: ventilation; BF: breathing frequency; VO_2_: oxygen consumption; VCO_2_: carbon dioxide production; RER: respiratory exchange ratio; HR: heart rate; EE: energy expenditure; PR: perceived readiness; EFP: early-follicular phase; LFP: late-follicular phase; MLP: mid-luteal phase. ^β^ Significant differences in LFP compared to EFP and MLP.

**Table 1 ijerph-18-00860-t001:** Sex hormone concentrations (Mean ± SD) on the testing days.

	EFP	LFP	MLP	c^2^	*p*
LH (mUI/mL)	7.27 ± 3.91	12.56 ± 8.29	5.96 ± 3.26	9.810	0.007 ^1^
FSH (mUI/mL)	9.14 ± 8.49	6.17 ± 2.95	3.44 ± 1.53	30.095	<0.001 ^2^
E2 (pg/mL)	38.78 ± 30.39	186.67 ± 154.56	138.11 ± 71.99	25.810	<0.001 ^3^
Progesterone (ng/mL)	0.33 ± 0.19	0.75 ± 1.79	11.99 ± 5.37	27.494	<0.001 ^4^
E2/progesterone ratio	0.15 ± 0.17	0.53 ± 0.54	0.03 ± 0.08	26.571	<0.001 ^5^

EFP: early-follicular phase; LFP: late-follicular phase; MLP: mid-luteal phase; LH: luteinizing hormone; FSH: follicle-stimulating hormone; E2: 17β-estradiol. ^1^ Significant differences in LFP compared to MLP (*p* = 0.006, d = 1.08, CI = 0.41 to 1.75). ^2^ Significant differences in MLP compared to EFP (*p* < 0.001, d = 2.76, CI = 1.83 to 3.69) and LFP (*p* < 0.001, d = 1.58, CI = 0.89 to 2.27). ^3^ Significant differences in EFP compared to LFP (*p* < 0.001, d = 1.79, CI = 0.99 to 2.59) and MLP (*p* < 0.001, d = 1.91, CI = 1.16 to 2.66). ^4^ Significant differences in MLP compared to EFP (*p* < 0.001, d = 1.79, CI = 0.97 to 2.61) and LFP (*p* < 0.001, d = 1.91, CI = 1.08 to 2.74). ^5^ Significant differences in MLP compared to EFP (*p* = 0.004, d = 1.16, CI = 0.44 to 1.89) and LFP (*p* < 0.001, d = 2.54, CI = 1.51 to 3.57).

**Table 2 ijerph-18-00860-t002:** Performance variables throughout the warm-up across the menstrual cycle phases.

	EFP	LFP	MLP	c^2^	*p*
V̇e (L/min)	48.2 ± 8.7	46.7 ± 8.3	49.0 ± 8.5	13.900	0.001 ^β^
BF (breaths/min)	32.6 ± 6.4	32.7 ± 5.6	33.3 ± 5.9	1.200	0.549
V̇O_2_/Kg (mL/kg/min)	29.1 ± 2.6	28.0 ± 2.3	28.5 ± 2.3	8.100	0.017 ^#^
V̇CO_2_ (mL/min)	1481.2 ± 215.6	1426.7 ± 202.4	1443.7 ± 189.8	3.600	0.165
RER	0.88 ± 0.05	0.88 ± 0.05	0.88 ± 0.05	0.105	0.949
HR (bpm)	136.0 ± 12.8	136.6 ± 12.2	136.1 ± 16.2	2.923	0.232
EE (Kcal/day)	11834.8 ± 1521.8	11377.7 ± 1598.5	11571.3 ± 1574.3	9.100	0.011 ^#^
RPE	9.3 ± 1.8	9.3 ± 2.0	9.5 ± 2.3	0.847	0.655
PR	4.9 ± 0.3	4.6 ± 0.5	4.6 ± 0.5	8.970	0.011

EFP: early-follicular phase; LFP: late-follicular phase; MLP: mid-luteal phase; V̇e: ventilation; BF: breathing frequency; VO_2_: oxygen consumption; VCO_2_: carbon dioxide production; RER: respiratory exchange ratio; HR: heart rate; EE: energy expenditure; RPE: rate of perceived exertion; PR: perceived readiness. ^β^ Significant differences in LFP compared to EFP and MLP. ^#^ Significant differences in LFP compared to EFP.

**Table 3 ijerph-18-00860-t003:** Performance variables throughout the cool down across the menstrual cycle phases.

	EFP	LFP	MLP	c^2^	*p*
Ve (L/min)	43.2 ± 6.4	42.6 ± 6.1	45.9 ± 6.0	10.048	0.007 ^£^
BF (breaths/min)	37.2 ± 6.5	37.7 ± 6.4	38.9 ± 6.1	6.723	0.035 ^γ^
VO_2_/Kg (mL/kg/min)	19.5 ± 2.7	19.0 ± 2.5	1975 ± 2.0	3.900	0.142
VCO_2_ (mL/min)	1069.4 ± 180.8	1058.4 ± 164.9	1109.6 ± 132.0	9.300	0.010 ^£^
RER	0.94 ± 0.06	0.97 ± 0.08	0.97 ± 0.07	2.947	0.229
HR (bpm)	137.9 ± 15.2	138.5 ± 13.6	137.3 ± 13.6	0.824	0.662
EE (Kcal/day)	8046.8 ± 1304.3	7865.7 ± 1390.3	8136.7 ± 1117.6	7.300	0.026 ^£^
RPE	9.8 ± 2.9	9.4 ± 2.0	10.0 ± 2.5	2.596	0.273
PR	4.1 ± 1.1	4.3 ± 0.7	4.0 ± 1.0	5.056	0.080

EFP: early-follicular phase; LFP: late-follicular phase; MLP: mid-luteal phase; Ve: ventilation; BF: breath frequency; VO_2_: oxygen consumption; VCO_2_: carbon dioxide production; RER: respiratory exchange ratio; HR: heart rate; EE: energy expenditure; RPE: rate of perceived exertion; PR: perceived readiness. ^£^ Significant differences in LFP compared to MLP. ^γ^ Significant differences in EFP compared to MLP.

## Data Availability

Not applicable.

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
