# Peer review of "Menstrual Cycle Phases Influence on Cardiorespiratory Response to Exercise in Endurance-Trained Females"

_ijerph, 2021, doi:10.3390/ijerph18030860_

Round 1
Reviewer 1 Report
Dear Authors,
Manuscript well-written, topic interesting and is an important one – the influence of menstrual cycle in Cardiorespiratory response. However, the manuscript has some points to be improved:
The manuscript has many abbreviations, which makes it difficult to read. In my opinion, you should use abbreviations only in the essential words in the text.
In some places in the text, you used two places after the period. Please standardize in just 1 place after the period (e.g. 1.5).
On the tables, please try to demonstrate where the difference between the groups is. So it would be easier to view, instead of reading the footer.
In general, the text has good English, however, at times it seems to be repetitive. Please see where you can remove duplicate information.
Author Response
Dear Authors,
Manuscript well-written, topic interesting and is an important one – the influence of menstrual cycle in Cardiorespiratory response. However, the manuscript has some points to be improved:
The manuscript has many abbreviations, which makes it difficult to read. In my opinion, you should use abbreviations only in the essential words in the text.
Thank you very much for your comment. Although it is true that so many abbreviations have been used, all of them are well known and previously used in literature. Nonetheless, some unnecessary abbreviations have been removed (MC, vVO2peak, FFA and AMPK) in order to clarify the manuscript. If some other abbreviations need to be removed, do not hesitate to let me know.
In some places in the text, you used two places after the period. Please standardize in just 1 place after the period (e.g. 1.5).
Thank you very much for noticing this edition mistakes. The journal has edited all the manuscript and it is prepared to be published, just in case it is accepted. Thus, these mistakes have been addressed with the journal style and edition.
On the tables, please try to demonstrate where the difference between the groups is. So it would be easier to view, instead of reading the footer.
Thank you very much for this suggestion. You are absolutely right; tables were hard to understand by reading the footer. What we have done is:
- When there is a difference between 2 menstrual cycle phases, we put in bold both numbers.
- When there is a difference in 1 menstrual cycle phase compared to the other two phases, I put in bold the one which is different.
From our point of view, this has been the best way to show where the differences are. However, we are not sure if it will be clear enough for readers. Hence, your feedback will be appreciated. If you have any other suggestion to demonstrate the Tables in a clearer way, please do not hesitate to tell us.
In general, the text has good English, however, at times it seems to be repetitive. Please see where you can remove duplicate information.
Thank you very much for your comment regarding the good English as well as for noticing this repetitive information. After rereading the manuscript, it is true that the discussion had some duplicate information which have been removed, Lines 329-332.
Finally, I would like to thank you for all your interesting suggestions and corrections that have greatly enhanced this work. I do appreciate them, and I wish my corrections addressed all the issues mentioned. Nonetheless, if you consider there is still something that needs to be addressed, please do not hesitate to tell me and I will work on it.

Reviewer 2 Report
This is an interesting study about the menstrual cycle effect on cardiorespiratory response to exercise. This is an important topic. However there are some major concerns. Comments below:
L65. Misses this study hypothesis.
As much as possible avoid to use abbreviations. There are too many and makes the text hard to read.
L282. Include the hypothesis and if it was confirmed or rejected.
L287-288. p values do not present tendecies. Peharps effect sizes does (if p is near 0.05 you can not assume such as "almost significant"). Please revise this afformation along the discussion.
The authors failed to present the study limitations and that will contribute to change the discussion. The ethnicity or race may affect the study outcomes. Daily habits, stress, motivation levels and so on... The authors may consider the uncontrolled variables and rewrite the discussion.i
The scussion section has too much theory and the results lacks of explanation and comparisons with literature.
Author Response
This is an interesting study about the menstrual cycle effect on cardiorespiratory response to exercise. This is an important topic. However, there are some major concerns. Comments below:
L65. Misses this study hypothesis.
Thank you very much for this comment. You are absolutely right, we missed to write the hypothesis. Now it has been included in line 67 as well as at the beginning of the discussion.
As much as possible avoid using abbreviations. There are too many and makes the text hard to read.
Thank you very much for your comment. Although it is true that so many abbreviations have been used, all of them are well known and previously used in literature. Nonetheless, some unnecessary abbreviations have been removed (MC, vVO2peak, FFA and AMPK) in order to clarify the manuscript. If some other abbreviations need to be removed, do not hesitate to let me know.
L282. Include the hypothesis and if it was confirmed or rejected.
Thank you very much for this suggestion, I do appreciate it. At the beginning of the discussion, in lines 294-299, the hypothesis and its confirmation have been included.
L287-288. p values do not present tendecies. Peharps effect sizes does (if p is near 0.05 you can not assume such as "almost significant"). Please revise this afformation along the discussion.
Thank you very much for noticing this big mistake. It has been addressed along the discussion (line 304 and line 325).
The authors failed to present the study limitations and that will contribute to change the discussion. The ethnicity or race may affect the study outcomes. Daily habits, stress, motivation levels and so on... The authors may consider the uncontrolled variables and rewrite the discussion.
Definitely. There are several variables we did not take into consideration when performing this study. Thank you so much for noticing it. We have included them in the limitation section at the end of the discussion, lines 371-372.
The discussion section has too much theory and the results lacks of explanation and comparisons with literature.
Thank you so much for this comment. On the one hand, it is true that the discussion has too much theory; maybe it is because I can not avoid talking about what I like the most….human physiology. On the other hand, there are not so many study research carried out with trained females evaluating cardiorespiratory response to a high intensity training programs regarding their menstrual cycle phases. Although I tried to include all previous research evaluating females´ cardiorespiratory system across the menstrual cycle, for some variables comparisons with previous literature have been hard to find. Consequently, we have used some theorical concepts in order to justify some of our findings.
However, I you still consider that there are some theorical concepts that are not essential please let me know and I will remove them from the discussion. Or it may be placed in the Introduction section.
Finally, I would like to thank you for all your interesting suggestions and corrections that have greatly enhanced this work. I do appreciate them, and I wish my corrections addressed all the issues mentioned. Nonetheless, if you consider there is still something that needs to be addressed, please do not hesitate to tell me and I will work on it.

Round 2
Reviewer 2 Report
The authors have made a great effort to improve the manuscript based on the reviewers commentaries.
This reviewer believes that this manuscript is now suitable for publication in IJERPH.